# On the Localization of Persistent Currents Due to Trapped Magnetic Flux at the Stacking Faults of Graphite at Room Temperature

**DOI:** 10.3390/ma15103422

**Published:** 2022-05-10

**Authors:** Regina Ariskina, Markus Stiller, Christian E. Precker, Winfried Böhlmann, Pablo D. Esquinazi

**Affiliations:** 1Division of Superconductivity and Magnetism, Felix-Bloch Institute for Solid State Physics, University of Leipzig, 04103 Leipzig, Germany; regina.ariskina@physik.uni-leipzig.de (R.A.); bohlmann@physik.uni-leipzig.de (W.B.); 2Zollsoft Praxissoftware GmbH, Ernst-Haeckel-Platz 5/6, 07745 Jena, Germany; markus@mstiller.org; 3Artificial Intelligence and Data Analytics Laboratory, Smart Systems and Smart Manufacturing, AIMEN Technology Centre, PI. Cataboi, 36418 Pontevedra, Spain; ceprecker@gmail.com

**Keywords:** defect-induced superconductivity, graphite, stacking faults, magnetic force microscopy

## Abstract

Granular superconductivity at high temperatures in graphite can emerge at certain two-dimensional (2D) stacking faults (SFs) between regions with twisted (around the c-axis) or untwisted crystalline regions with Bernal (ABA…) and/or rhombohedral (ABCABCA…) stacking order. One way to observe experimentally such 2D superconductivity is to measure the frozen magnetic flux produced by a permanent current loop that remains after removing an external magnetic field applied normal to the SFs. Magnetic force microscopy was used to localize and characterize such a permanent current path found in one natural graphite sample out of ∼50 measured graphite samples of different origins. The position of the current path drifts with time and roughly follows a logarithmic time dependence similar to the one for flux creep in type II superconductors. We demonstrate that a ≃10 nm deep scratch on the sample surface at the position of the current path causes a change in its location. A further scratch was enough to irreversibly destroy the remanent state of the sample at room temperature. Our studies clarify some of the reasons for the difficulties of finding a trapped flux in a remanent state at room temperature in graphite samples with SFs.

## 1. Introduction

Defect-induced superconductivity (DIS) is a remarkable phenomenon in solid state physics that triggers superconductivity in certain regions of the atomic lattice. For example, the one predicted in topological flat-band systems [1] like at the surface of rhombohedral (3R) graphite [2,3,4], at the stacking faults or interfaces between Bernal (2H) and 3R stacking orders [3,5,6], or between multilayer graphene twisted stacking order regions [7,8,9]. DIS is also observed and/or predicted in semiconducting superlattices and ultra thin films [10,11,12], at the interfaces of pure Bi and BiSb bicrystals  [13,14,15], and at certain interfaces of semiconducting superlattices with [16] or without strain [17,18].

Carbon-based materials belong to the promising materials for high-$T_{c}$ superconductivity (HTS) at normal pressure. Josephson tunneling-like behavior has already been reported in 1974 in a disordered graphite powder [19,20], a report considered to be the first hint for room-temperature superconductivity (RTS) though not recognized thoroughly in the community, probably due to the difficulties to reproduce those results. Successive transport and magnetic studies on graphite bulk samples [21,22,23,24], graphite powders [25,26], and transmission electron microscope (TEM) graphite lamellae [27,28], provide further hints for the existence of RTS at certain interfaces or stacking faults (SFs) embedded in the graphite matrix, with a maximum critical temperature of $T_{c}∼350\;$K suggested by transport, magnetization [23], and MFM [24] measurements. A possible origin for the HTS in graphite-based systems is thought to be related to flat band regions localized at certain SFs of the graphite lattice. A flat band, a dispersionless energy relation in *k*-space in the vicinity of the Fermi level, may occur at the surface of 3R graphite, at the interfaces or SFs between twisted single 3R or 2H crystalline phases, or between untwisted 3R and 2H phases. Indeed, flat band-related enhancement of the electronic density of states has been observed experimentally at the surface of 3R graphite [29,30,31,32]. Similarly, van-Hove singularities at certain regions of twisted bilayer graphene have been reported [33], which can be related to the superconductivity in twisted bilayer or 3R-trilayer graphene with critical temperature $T_{c}≲5$ K [4,8].

### Assumptions and Experimental Restrictions to Observe Room Temperature Superconductivity in Graphite

Assuming that RTS can be localized at certain SFs in the graphite structure, it is not straightforward to experimentally verify its existence due to the following reasons:

(1) Experimental studies suggest that RTS in graphene-based systems might be found at the SFs between 3R and 2H stacking orders [23,24]. Relatively expanded regions with 3R stacking order are found in highly oriented pyrolytic graphite (HOPG) as well as in natural graphite crystals, reaching a relative concentration ≲10% in HOPG and as large as ∼25% in natural graphite [23,34]. The amount and the localization of the 3R stacking order region strongly depend on each graphite sample, for recent examples see [35,36]. It means that reproducibility is not easy to achieve even in samples of the same batch. Because it is not yet possible to produce systematically 3R regions with well-defined interfaces to 2H regions, we are restricted to use bulk graphite samples (natural or HOPG) with a sufficiently good crystalline order.

(2) Experimental studies of the electrical resistance and current-voltage (I−V) characteristic curves of graphite TEM lamellae (where an electrical contact at the edges of the SFs is possible) indicate that the uppermost critical temperature $T_{c}$ (defined as the temperature where the resistance shows a maximum) increases the larger the area of SFs [37]. This kind of size-dependent effect of $T_{c}$ has been already reported for superconducting-metal (Nb/Al) multilayers (leaving the thickness of each of the layers constant) [38]. Its origin has been tentatively given based on weak localization corrections to $T_{c}$ for 2D superconductors [38,39,40]. Whatever the reason, the apparent size-dependence of $T_{c}$ reported in [37] would restrict the observation of RTS to graphite samples with SFs of area ≫10${}^{2}$ μm${}^{2}$.

(3) In SFs of large size we do not expect to have homogeneous superconductivity all over a specific SF, but a granular one, i.e.,  Josephson-coupled superconducting regions. Granular superconductivity behavior in the magnetization has been reported in water treated graphite powders [25] as well as in bulk macroscopic HOPG samples with SFs [22]. Transport measurements in bulk natural crystals [23] and I−V characteristic curves of TEM lamellae [27] also indicate the existence of Josephson-coupled (granular) superconducting regions. The response of granular superconductors to magnetic fields is more complicated than in homogeneous superconducting samples. A thorough description of the different stages one expects in the magnetization as a function of the applied field in a granular superconductor is given in [41]. Such behavior has indeed been observed in the systematic measurements reported in [22,25]. Even the reported transition in a twisted bilayer graphene mesoscopic sample [8] does not appear to behave as a homogeneous but as a granular superconductor, as a comparison between those results and the ones observed in graphite TEM lamellae indicates [42].

Fixing ideas, we show in Figure 1a a scanning electron microscope (SEM) image of a graphite bulk sample and a scanning transmission electron microscope (STEM) image taken from a graphite lamella with the electron beam applied parallel to the graphene interfaces in Figure 1b, where the *c*-axis is perpendicular to the graphene planes, SFs or interfaces. The SFs can be easily recognized in the TEM image as the boundaries between regions of different shades of grey. The sharper the contrast between the regions is, the better is the interface or SF defined. The SF can be between regions of different twist angles with respect to the common *c*-axis or between regions with different stacking orders. Note that the crystalline regions with a homogeneous grey color have different thicknesses. Most of the single crystalline regions have Bernal stacking order and a smaller amount, the thinnest ones, can have the rhombohedral one. Any determination of the twist angle or the stacking order via STEM needs an appropriate previously calibrated sample, which is not yet available.

Let us assume that we have Josephson-coupled superconducting regions as depicted in Figure 1b,d, i.e., granular superconductivity, embedded within the 2D SF. In such granular systems, we expect a trapped flux after removing an applied field $H>H_{c1}^{w}$. The critical field $H_{c1}^{w}$ is a sample dependent characteristic field above which flux lines and the associated currents penetrate the sample via the weakest links, as magnetization [41] and magnetoresistance, see, e.g., [43], measurements in granular high-temperature superconductors indicate.

Decreasing the applied field might give rise to a macroscopic current loop through Josephson coupled superconducting regions, as sketched in Figure 1b. The current loop generates a trapped magnetic flux within the loop, in a region where no shielding of the field occurs. Traces of a macroscopic current path have been detected in a natural graphite sample at remanence via magnetic force microscopy measurements [24]. Other experimental hints manifesting the existence of persistent currents at remanence are the magnetization measurements on bulk graphite [22,44], of finely dispersed HOPG grains [26] and of water-treated graphite powders [25].

The aim of this study is to reproduce the magnetic force microscopy (MFM) results reported in [24]. We need to understand the reasons for the difficulties one has in observing this permanent current path at room temperature in graphite samples. For example, at which depth from the sample surface should the current within the SF be located in order to get a reasonable large phase change of the MFM signal.

## 2. Magnetic Force Microscopy and Monopole Model Description

### 2.1. Magnetic Force Microscopy

Magnetic force microscopy has been successfully used for imaging pinned vortices at the surface of low-temperature superconductors [45,46]. MFM at room temperature has been used to investigate the magnetic properties of HOPG samples [47,48,49] together with the Kelvin probe force microscopy (KPFM) technique [50]. It also has the capability to identify and characterize large areas of trapped flux (>$10^{2}$μm${}^{2}$) in granular superconductors as well as to localize current lines flowing between weakly coupled superconducting regions in graphite [24]. In this case the magnetic field distribution of the trapped flux is imaged through the interaction at a certain distance between the ferromagnetic cantilever tip and the field gradient. The scan is done twice, where the second scan reproduces the topography configuration at a certain height, minimizing other non-magnetic forces acting on the tip.

The expression derived for the *z*-component of the force acting on the tip is given by [51]:(1)$$\frac{\partial F_{z}}{\partial z}=-\frac{q}{\mu_{0}}\frac{\partial B_{z}}{\partial z}+m_{x}\frac{\partial^{2}B_{x}}{\partial z^{2}}+m_{y}\frac{\partial^{2}B_{y}}{\partial z^{2}}+m_{z}\frac{\partial^{2}B_{z}}{\partial z^{2}},$$
where *q* is the magnetic monopole flux and $m_{i}$ is the magnetic moment of the tip in the *i* = *x, y, z* directions. The field gradient interacting with the magnetic moment of the tip produces a small change in the cantilever resonance frequency, which translates in a phase change. This phase change can be derived as [51]:(2)$$\varphi =-\frac{Q}{k}\left(\frac{\partial F}{\partial z}\right),$$
where *Q* is the tip quality factor and *k* its spring constant.

### 2.2. Cantilever Tip Approximation

For some tip materials and upon application, the tip can be simulated either as a magnetic monopole with the force $F_{z}=qB_{z}$ or as a magnetic dipole with the force $F_{z}\partial B_{i}/\partial z$. Our experimental MFM data can be modeled using the monopole tip approximation. Assuming that the cantilever is only magnetized in the z-direction, the force derivative originated from a current line source can be written as [52]:(3)$$\frac{\partial F_{z}}{\partial z}=\frac{qI}{\pi }\frac{xz}{{\left(x^{2}+z^{2}\right)}^{2}},$$
where *I* is the current of the line source and *z* the scan height to the current path. The phase can be then modeled using Equations (2) and (3) as:(4)$$\varphi =-\frac{Q}{k}\frac{A}{\pi }\frac{xz}{{\left(x^{2}+z^{2}\right)}^{2}}.$$
where *A* = *qI*.

## 3. Experimental Setup, Sample Preparation, and Precharacterization

Initially, the surface of more than fifty well-ordered graphite samples (taken from bulk HOPG and natural graphite) were cleaned and mechanically exfoliated. The maximum concentration of contaminants in the selected graphite samples, measured by RBS/PIXE, is ≲20 ppm [23]. All samples were fixed on a nonmagnetic substrate and electrically grounded. We have brought the samples to a remanent state by applying an external magnetic field using either a homemade electromagnet system or a permanent magnet. The applied field of the permanent magnet measured was 1.5 kOe and was measured with a Hall sensor at the position of the sample. The field provided by the electromagnet reached a maximum of 0.5 kOe. The field was applied always perpendicular to the SFs and the graphene planes of the samples.

The magnetic force microscope used to identify the permanent current path at room temperature is a Bruker Dimension Icon Scanning Probe Microscope. The  MFM measurements were performed with a combination of two modes: PeakForce Tapping Mode and Lift Mode. The Lift Mode was operated at a constant height of 100 nm. The tips used were magnetic-coated MESP-HM-V2 with a magnetic moment of 3 × $10^{-13}$ emu, medium nominal coercivity of 400 Oe, and nominal tip radius of 80 nm. The MFM tips were similar to those used in [24], which results as a function of the distance to the sample surface, indicating that the field produced by the MFM FM tip does not influence the current loop. All the MFM measurements presented in this study were done at zero applied magnetic field.

Before applying the magnetic field, we have scanned the sample’s surface with our MFM to check whether magnetic signals were observed in the virgin state of the samples at zero applied field. In agreement with the study reported in [24], apart from spurious artifacts in the phase related to certain topography features at the sample surface, we did not observe any signal comparable with the one we were looking for and compatible with a current line. We were not able to recognize any magnetic domains in any of the clean samples that can be interpreted in terms of ferromagnetic regions.

We have spent more than 1.5 years searching for the expected phase signals in the remanent state (after applying and removing the magnetic field) of a large number of graphite samples without success. Finally, we found a natural graphite sample from Sri-Lanka (of dimension ∼2000μm ×300μm ×200μm, similar to those reported in [23,24]). After we present and discuss the obtained results, it will become clear why it is not surprising to have such a low success rate in finding a permanent current path via MFM. One possible precharacterization of the graphite sample, prior to the MFM measurements, is to check whether there is an irreversible field behavior in the magnetoresistance.

In case a graphite sample has regions where a magnetic flux is trapped in a remanent state, its magnetoresistance should show a hysteresis after increasing and decreasing the field. The reason for a field hysteresis is related to the relatively large sensitivity of the electrical resistance of graphite samples to magnetic fields even at room temperature, which is related to the existence of SFs [53]. Figure 2 shows the field dependence of the relative change of the electrical resistance, i.e., the magnetoresistance, of a natural graphite sample taken from the same batch. We observe that the resistance clearly increases in field with a negative curvature, similar to that reported in [23]. If we stop the field sweep at a certain field and return to zero, i.e., to a remanent state, the resistance remains within the relatively short measuring time (for the time dependence of the resistance in the remanent state see, e.g., Figure 21 in [23]) indicating that a certain amount of magnetic flux has been trapped. There are actually only two possible origins for field irreversibility in magnetoresistance. Namely, either pinning of domain walls as in ferromagnetically ordered systems, or pinning of vortices or fluxons in superconducting regions. Because we do not have any evidence for the existence of magnetically ordered regions in the graphite samples, the field hysteresis cannot be due to domain-wall pinning of magnetic domains. Therefore, superconductivity appears to be the only possible reason for the field hysteresis in the magnetoresistance and the trapped magnetic flux.

We would like to emphasize that the field hysteresis in the magnetoresistance is a hint to finding with MFM a flux trapped region. The ability to localize it depends obviously on the distance between the permanent current path and the MFM tip.

## 4. Results and Discussion

### 4.1. The MFM Phase Signal at the Current Path

We assume that a permanent current loop maintains a flux trapped somewhere in the sample, localized at a 2D superconducting SF. We expect that the current path is not a one-dimensional line but a 2D path area. Its width and therefore its current density depend on the SF properties and on the location within the sample. On the other hand, the topography of the sample surface is far away from being flat. As sketched in Figure 1d, it means that the distance between the MFM tip and the SF would depend on the location of the tip with respect to the sample. Therefore, and for simplicity, we will not take into account explicitly a finite broadening of the current path, but instead, we will simulate the phase change at the current path location by changing slightly the height *z* of the tip from the surface in Equation (Equation 4).

Figure 3a shows an MFM phase image in a region around the current path clearly recognized as the border line between two different color regions. This difference in color is due to the difference in the field gradient, related to the absolute value and direction of the field vector in the *z*-direction. In general, it took us several weeks to a couple of months of measurements at different sample positions to find the current path region.

Figure 3b shows the line scan through the path (black arrow in (a), black line in (b)) and the results of the fit to Equation (Equation 4) (dashed red line). We found that, in general, the experimental line-scan data can be well fitted by the simple one-dimensional relation given by Equation (Equation 4). The fit in Figure 3b was obtained using a spring constant k=3±0.06 N/m, a scan height z=100±5nm, a quality factor of the tip Q=277±8, taken from [55]. The constant $A=\left(10.5\pm 0.45\right)\times 10^{-20}$ CA was adjusted to fit the experimental curve.

### 4.2. Influence of the Applied Magnetic Field on the Remanent Current Value

The first observation of a current path after applying and removing the ∼500 Oe field was after nearly two weeks of continuous MFM measurements through the sample. The found MFM phase image in Figure 4a and the current path with the corresponding line scan shown in Figure 4c indicate a discernible current path but of relatively small phase difference. Therefore, we decided to apply a field of 1.5 kOe to the sample and measure again the current path within the same region in a remanent state. The phase signal image measured in the remanent state, see Figure 4b and the line scan in (d), show that the phase difference between the two field regions separated by the current path increased by a factor of two. With the monopole tip approximation a reasonable good fit of the line scans was achieved (dashed red lines in (c) and (d)) taken a two times larger scan height, i.e., 200 nm for both cases. In this case, the constant $A=\left(24\pm 1\right)\times 10^{-19}$ CA after applying 500 Oe (c) and $\left(11.9\pm 0.5\right)\times 10^{-17}$ CA after 1.5 kOe (d). The difference in A=qI indicates that the value of the current *I* became larger after applying a larger field.

We note that the signal-to-noise ratio between the line scan in Figure 3b and those in Figure 4c,d are similar, in spite of nearly two orders of magnitude difference in the phase difference $\Delta \varphi_{2}$. Let us assume that the difference in the magnitude of $\Delta \varphi_{2}$ is mainly due to the difference in the height between the MFM tip and the interface position. In this case this apparent constancy of the noise to signal ratio would indicate that the origin of at least part of the “noise“ originates at the same 2D interface. Evidently, more experiments are necessary to check for the origin of this rather unusual behavior.

Due to the uneven topography of the relatively large graphite sample surface, it is quite difficult and measuring-time intensive to get an image of the whole current loop using MFM. In the measured sample we could follow the current path up to the edges of the loop located near the two opposite ends of the sample at a distance of ∼1.6 mm, see Figure 5.

### 4.3. How Deep Is the SF of Interest?

One important open question is the typical distance of the SF of interest from the sample surface that would provide a large enough MFM signal. If this distance is larger than the typical distance between SF (of the order of 100 nm to 200 nm in HOPG samples [27]), then we would expect to measure MFM signals in most large samples and samples’ areas. Because this is not compatible with our experience, we assume that the SFs of interest should not be too deep inside the sample.

To get some knowledge on this issue and after several months of measurement of MFM phase images around the current path, we have “produced” a ∼10 nm deep scratch at the current path with the MFM tip. The MFM phase image of the region, before the scratch, is shown in Figure 6a. The scratch was located at the upper part of that image, where the black arrow is shown in Figure 6b. After the scratch and without changing the remanent state of the sample, we observed that the current path shifted its location expanding the loop area, see Figure 6b. If we assume that the total magnetic flux should remain constant after the scratch, then an expansion of the loop area should be accompanied by an effective decrease in the current amplitude. However, differences in the surface topography at the current paths do not allow a quantitative comparison of the current amplitudes before and after the scratch. More systematic evidence is necessary to understand the phenomenon.

The obtained result indicates that the scratch affected the superconducting grains and/or the Josephson coupling between them at the SF of interest. Figure 6c shows the topography scan through the scratch region indicating a depth of the order of 10 nm. We assume therefore that the SF of interest was at ≲10 nm from the nominal sample surface. This result also indicates that, taking into account the phase sensitivity of the MFM, it appears to be rather difficult to localize a current path of a superconducting SF located much deeper.

We note that this kind of “destructive” experiment has some risks. A second deep scratch produced at another region a few months later caused the trapped flux and the current path to vanish. After applying a magnetic field several times to trigger a remanent state again we were not able to find the current path again.

### 4.4. Estimate of the Absolute Value of the Current

To estimate the value of the current, first, we measured the phase signal using a 2D-like current loop (≃11 μm diameter, 200 nm thickness, and w≃1.5μm width) made of a Au film deposited on a dielectric substrate, as shown in [24]. The phase shift $\Delta \varphi_{1}$ between the value at the center and outside the loop as well as the total phase shift at the position of the current path $\Delta \varphi_{2}$, see Figure 7a, were measured as a function of the applied current as well as a function of the scan height at 10 mA applied current. Taking into account those results we obtain the phase shift $\Delta \varphi_{2}$ as a function of the applied current at 100 nm scan height shown in Figure 7b.

A first rough estimate of the permanent current $I_{p}$ triggered after removing the applied field in our sample, can be done assuming $I_{p}∼I\left(\Delta \varphi_{2}\right)\times r$, where $r=w_{1}/w_{2}$ the ratio of the widths of phase shift of the permanent current path ($w_{1}≲100$ nm) to the one of the Au current loop ($w_{2}≃1.5\;\mu$m). Our measurements of the graphite sample indicate a phase shift $0.2^{\circ }≲\Delta \varphi_{2}≲5^{\circ }$, from which we obtain a permanent current in the range 0.05 mA $≲I_{p}≲1.3\;$mA. These values are clearly larger than expected taking as reference the critical Josephson currents measured by transport contacting directly the edges of the SFs in [27]. However, more reasonable values are obtained if we take not only the width ratio of the paths but also the thickness ratio. At the SF we expect a current path thickness of ∼1 nm in comparison to 200 nm for the Au film. In this case, we have 0.2 μA $≲I_{p}≲6\;\mu$A. The results obtained in [27] indicate the Josephson critical current values between 55 nA and 5.5μA, see Figure 5 in that publication.

Let us now estimate the value of the field inside the loop. According to the Biot–Savart law $B=\mu_{0}I/2R$, where *R* is the radius of a circular loop. We assume that the radius of a circular loop having an area similar to the loop ellipse (in our case, large radius of ellipse ∼1.7 mm, small radius ∼100 μm) is equal to 0.4 mm. The magnetic field produced by this circular loop is then ≲7.7 nT, too small to be measured with conventional Hall sensors.

### 4.5. Flux Creep

To further verify the superconducting roots of the phenomenon, the time dependence of the position of the current path was investigated. In [24] the time dependence of the current through the path $I_{p}\left(t\right)$ was obtained by measuring the total phase shift $\Delta \varphi_{2}$, see Figure 3, at different times. The obtained result followed the typical logarithmic time dependence expected for flux creep in superconductors. The relative change of $I_{p}\left(t\right)$ was very small, of the order of 0.5% in two days of measurements and 22 days after removing the applied field. This last can be taken as a proof of the permanence of the triggered current $I_{p}$ and the negligibly small resistance value [24].

In this study, we decided to measure the time dependence of the current path in another way. Namely, by measuring the distance between an appropriate reference point and the position of the current path. The distance d(t) of the current path, see Figure 8a, was acquired in an area of 5×1μm${}^{2}$ within 1 day, 18 days after removing the applied field. The result is shown in Figure 8b. Similar logarithmic time dependence has been obtained in the magnetoresistance of a natural graphite after applying a fixed magnetic field at different temperatures [23], and in the magnetization of water-treated graphite powder where superconducting-like behavior has been also reported [25].

The shift with time of the current path in the graphite sample is related to the reduction of the area of the trapped flux, i.e., to a decrease of the absolute value of the magnetic moment m(t). Roughly speaking, we can assume that $m\left(t\right)\propto I\left(t\right)A\left(t\right)∼I\left(t\right){\left(L+d\left(t\right)\right)}^{2}$, where *L* is a fixed length from an effective center of the loop. This length is time independent and L≫d(t). Taking into account that the relative change of the current with time within one day is very small, the main time dependence of the magnetic moment would be m(t)∝d(t). Therefore, the relative change of the distance d(t) defined in Figure 8a would follow the flux creep relation according to:(5)$$d\left(t\right)/d\left(0\right)≃1-\left(k_{B}T/U_{a}\right)\ln\left(1+t/\tau \right),$$
where $k_{B}$ is the Boltzmann constant, T the temperature, $U_{a}$ is an effective activation energy, and τ a time constant characterizing the transition prior to the pure logarithmic time relaxation. Assuming as usual τ∼10 s and taking the absolute time from the day we removed the field, the change in d(t) can be roughly fitted with d(0)∼50μm and $k_{B}T/U_{a}∼0.08$, see Figure 8b, which means an effective activation energy $U_{a}∼10^{-19}$ J. This value is roughly two orders of magnitude smaller than the one estimated from the time dependence of the electrical resistance at 300 K, but at a fixed field of $10^{4}$ Oe, which probably is the main reason for the difference in $U_{a}$ [23]. The value of $k_{B}T/U_{a}$ obtained from the fit of d(t)/d(0) to Equation (Equation 5) is rather independent of our chosen zero time point. For example, if instead of taking the time elapsed after removing the field, we take the absolute time passed at each phase image after starting the measurements, we obtain a similar value for $U_{a}$ with a different value of d(0). A video on the MFM phase measurements during one day is available as Appendix A, see Appendix A below.

## 5. Conclusions

The investigation of the trapped magnetic flux through a persistent current after removing a given magnetic field applied to a large graphite sample is a crucial issue, necessary to characterize the granular room temperature superconductivity localized at certain stacking faults (or interfaces) existing in the graphite structure. The dimensions of these interfaces and their locations are difficult to specify, making it difficult to connect electrodes without damage, to measure one of the usual characterization proofs, i.e., a zero electrical resistance. Alternative methods, like measuring the Meissner effect are also problematic due to the two-dimensionality of the superconducting regions implying a demagnetizing factor near one. Therefore, the observation of permanent, non-dissipative currents directly with MFM (or with micro-Hall sensors) is one possible way to experimentally study the “hidden superconductivity” [56] in the graphite structure.

One of the aims of this study was to reproduce the magnetic force microscopy results reported in [24]. This aim was achieved. Moreover, our studies indicate that reasonable large phase signals can be obtained if the SF in question is not far away from the sample surface. Our scratch experiment provides a depth of the order of 10 nm. After measuring a large number of graphite samples, the possibility to measure a trapped flux at room temperature, localized at certain SFs, appears to be limited to graphite samples of large size, emphasizing the importance of the phenomenology on the size dependence of the apparent critical temperature published in [37]. Its implication is clear: similar MFM experiments have to be done at lower temperatures. The fact that both samples where we were able to measure a trapped flux with MFM at room temperature were natural graphite samples suggests that the existence of rhombohedral stacking order is of importance for room temperature superconductivity. Natural graphite crystals are still the samples with the largest rhombohedral fraction.

To conclude, we would like to note that, in general, it is difficult to keep an MFM system for months of non-interrupted, continuous measurements without crashes of the MFM tip on the sample surface, especially if the surface is uneven. As we have demonstrated in this study, having an interruption of the first triggered current path through a mechanically performed scratch does not necessarily mean that it (and the flux trapped) would completely vanish, but the path can find a new route if superconducting patches exist in the surroundings.   

## Figures and Tables

**Figure 1 materials-15-03422-f001:**
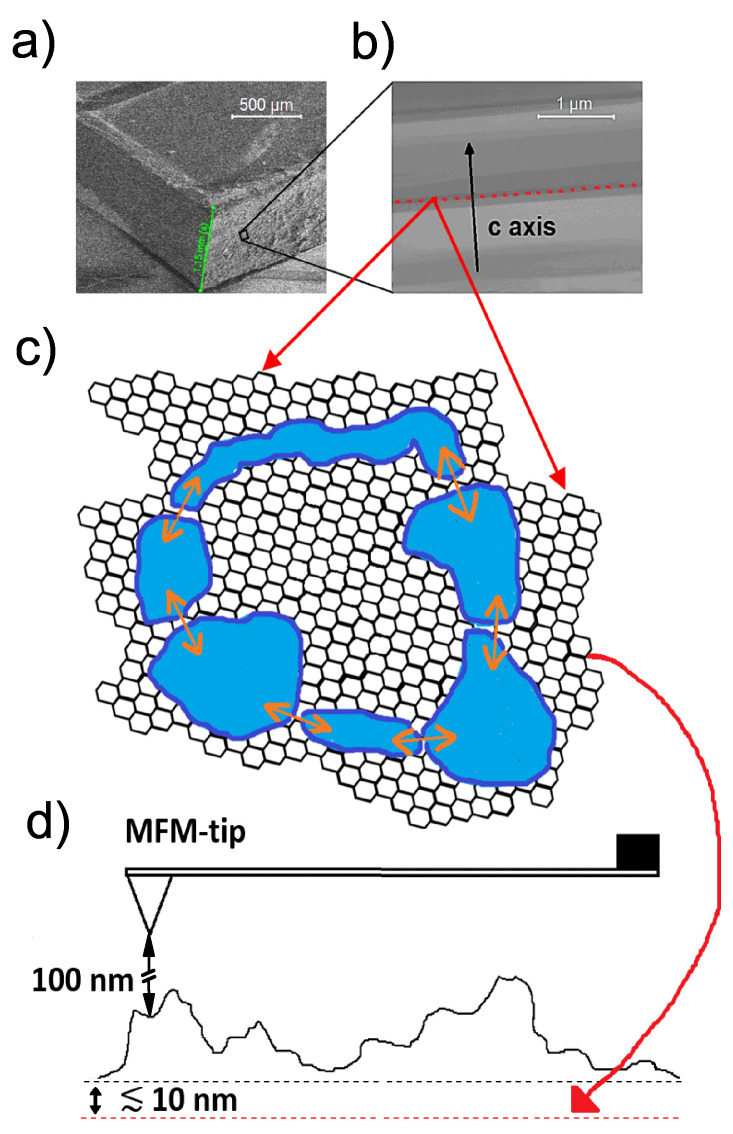
(**a**) Scanning electron microscope (SEM) of graphite bulk sample and (**b**) scanning transmission electron microscope (STEM) scan of a lamella taken parallel to the graphene interfaces, where the *c*-axis is perpendicular to the crystalline planes. (**c**) Sketch of a 2D interface at an SF with embedded superconducting patches (blue regions) and Josephson current (orange arrows) circulating between them. (**d**) Sketch of the topography of a graphite sample with a superconducting SF (dashed red line) near the sample nominal main surface (dashed black line). The distance of the SF to the MFM tip depends on the position within the sample area, a fact that affects the absolute value of the measured phase change at the current path location.

**Figure 2 materials-15-03422-f002:**
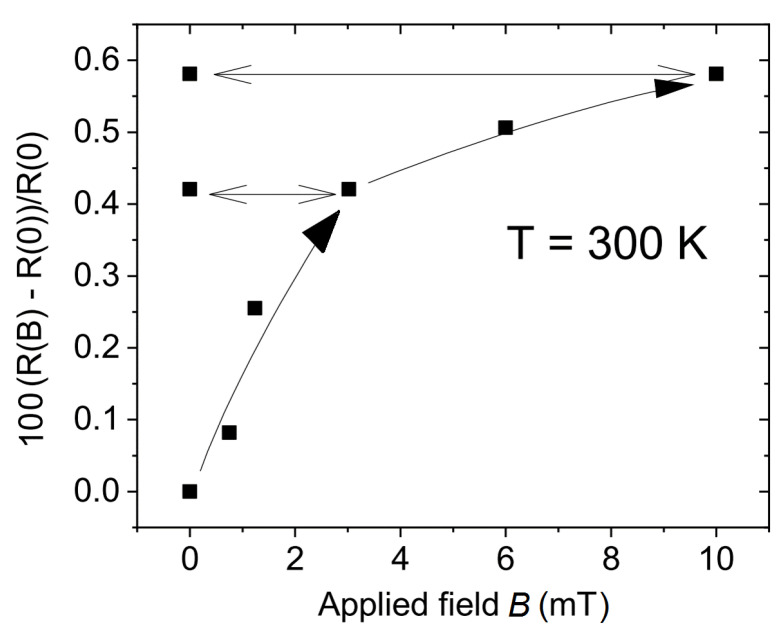
Field dependence of the normalized resistance at 300 K of a natural graphite sample from the same batch as the one used for MFM. Black squares represent the experimental data and the lines are only a guide for the eye. The field is applied perpendicular to the SFs and graphene planes of the sample. The arrows indicate the sweep field direction. Note that the values of the magnetoresistance reach ∼0.6% at 10 mT. With the LR700 resistance bridge we have used, the relative error of each of the resistance points is ≲$2\times 10^{-4}$ R(0). Similar results were reported in [23].

**Figure 3 materials-15-03422-f003:**
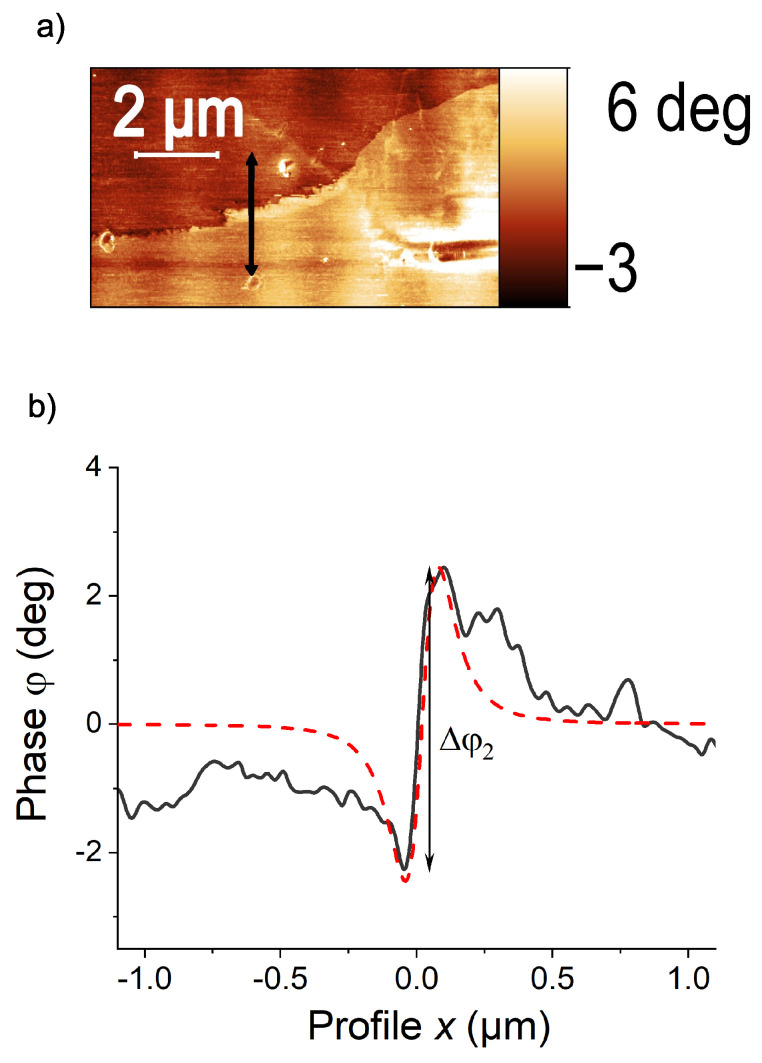
(**a**) MFM image of a sample region with the trapped flux. The current path shows a typical meander-like shape [24,54]. (**b**) Scan line (black) and MFM model (red dash line) using the point monopole tip approximation, see Equation (Equation 4). The vertical arrow indicates the definition of phase difference $\Delta \varphi_{2}$.

**Figure 4 materials-15-03422-f004:**
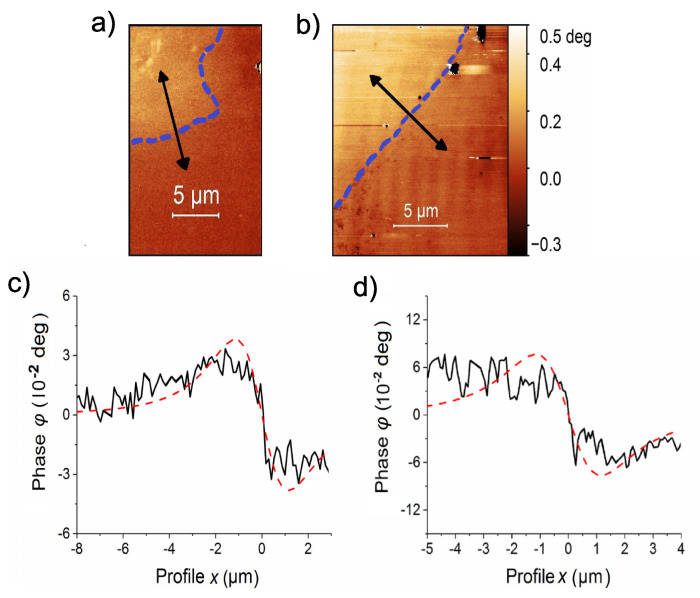
Phase images at the same position of the sample in remanent state (zero applied field) after applying magnetic fields of 500 Oe (**a**) and 1.5 kOe (**b**). Figures (**c**,**d**) show the corresponding profile lines. The dashed red lines are fitting curves. The fitting was performed according to the monopole tip approximation with the constant $A=\left(24\pm 1\right)\times 10^{-19}$ CA for (**c**) and $\left(11.9\pm 0.5\right)\times 10^{-17}$ CA for (**d**).

**Figure 5 materials-15-03422-f005:**
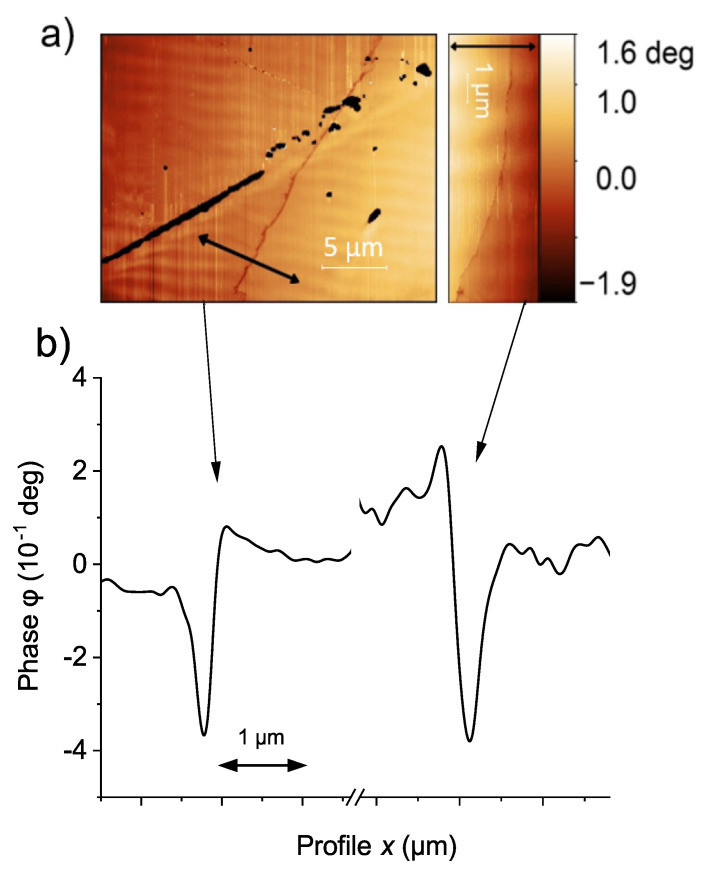
(**a**) MFM scan phase images of the sample at two locations at the opposite ends of the sample near its edges. (**b**) Line scans taken at the black arrows in (**a**). The distance between the two positions is ∼1600 μm.

**Figure 6 materials-15-03422-f006:**
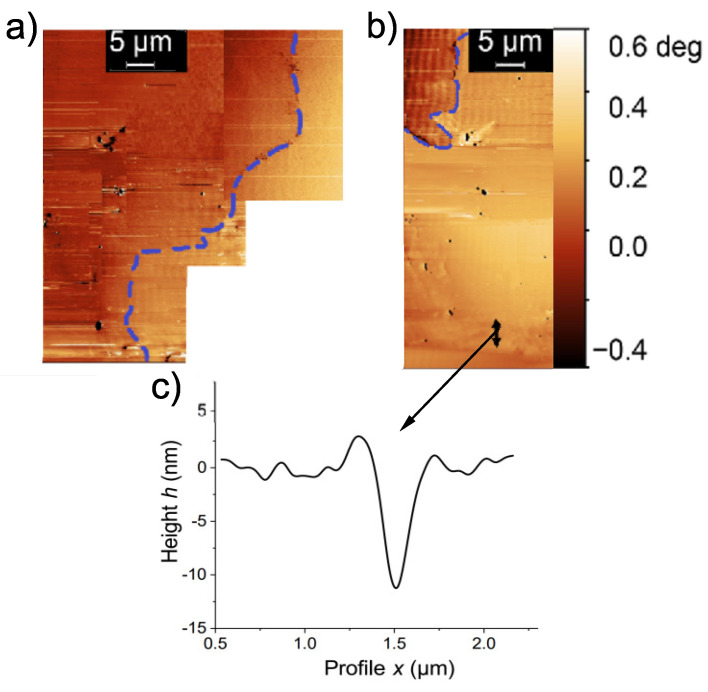
MFM phase images of a region around the current path (dashed blue line) before (**a**) and after the scratch (**b**). The black arrow in (**b**) marks the region of the scratch, which topography was measured with an AFM, see scan line profile in (**c**).

**Figure 7 materials-15-03422-f007:**
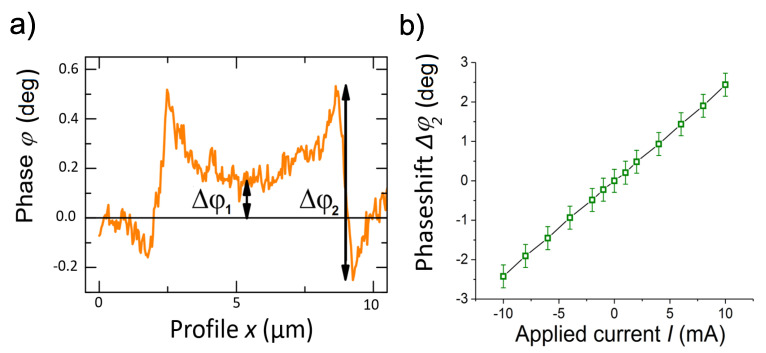
(**a**) Line scan of the phase through the current loop at 3 mA current and the definitions of the phase differences. (**b**) Phase difference $\Delta \varphi_{2}$ of the current loop as a function of the applied current at 100 nm lift scan height.

**Figure 8 materials-15-03422-f008:**
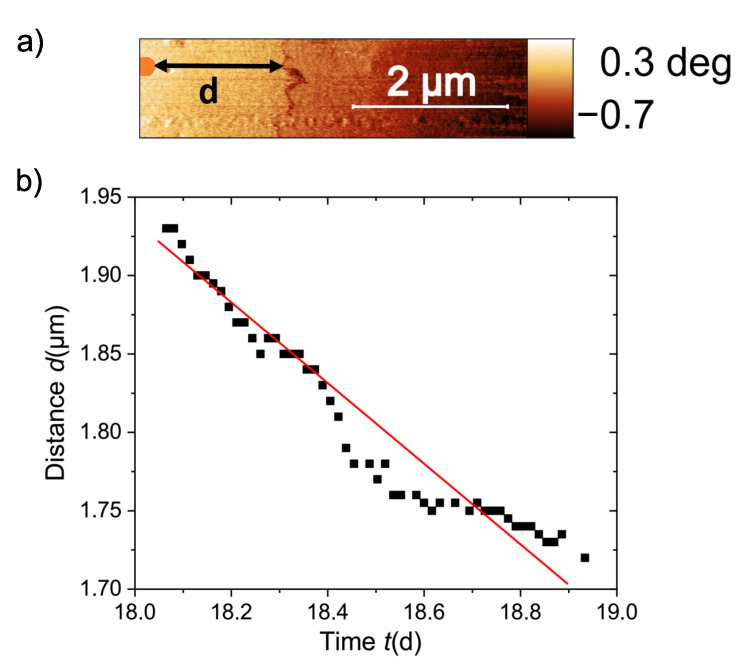
(**a**) Definition of the distance *d* between the reference topography point (orange disc at the left of the image) and the position of the current path delimited by the dark meander-like line. (**b**) Distance *d* vs. logarithm of time within a period of ≈24 h and starting the measurements 18 days after removing the applied field. The continuous line is the fit to Equation (Equation 5).

## Data Availability

Data are available from the corresponding author on request.

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
