# Peer review of "On the Localization of Persistent Currents Due to Trapped Magnetic Flux at the Stacking Faults of Graphite at Room Temperature"

_materials, 2022, doi:10.3390/ma15103422_

Round 1

Reviewer 1 Report

The manuscript presents the results of the authors' experimental efforts in a search for a granular room-temperature superconductivity manifestations in scanning magnetic-force microscopy. The authors discuss the routes and the difficulties in this kind of a study and suggest that the superconducting regions located at stacking faults can be revealed via the MFM phase contrast anomalies at the edges of the superconducting loops formed by the Josephson-connected patches. Only two cases out of about 50 studied samples revealed the detectable phase-contrast patterns, and these findings are described in the paper. In my opinion, the manuscript by Ariskina et al brings new useful information to the field of a long-expected discovery of the room-temperature superconductivity in carbon-based materials, it is clearly written and can be accepted for publication in the Materials journal after performing several minor corrections:

1) My biggest concern that could cause the reader's doubts in the presented results originates from a near-identical signal-to-noise ratio in the line scans presented in Fig.3(b) and in Figs.4(c) and (d) while the magnitude of the phase contrast in Fig.3(b) is almost two orders of magnitude larger than in Figs.4(c) and (d). The authors should address this unexpected discrepancy somehow.

2) In my opinion, the units in the designation of the y-axes in Figures 3(b), 4(c), 4(d), 5(c), 7(a) and 7(b) should be changed from the unconventional "o" symbol to "deg".

3) The values of the A parameter are presented in a wrong way. Thus, instead of A = 10.5 ± 0.45 × 10-20 (line 211), an appropriate way is A = (1.05 ± 0.05) × 10-19. This kind of unconventional number presentation is found in lines 211, 223 and 224, and two times in the caption to Figure 4.

4) In line 164 in the indication of the sample size a sign "×" is missing.

Author Response

Reply: We thank the reviewer for the positive comment on our Manuscript.

Reviewer wrote: 1) My biggest concern that could cause the reader's doubts in the presented results originates from a near-identical signal-to-noise ratio in the line scans presented in Fig.3(b) and in Figs.4(c) and (d) while the magnitude of the phase contrast in Fig.3(b) is almost two orders of magnitude larger than in Figs.4(c) and (d). The authors should address this unexpected discrepancy somehow.

Reply: We thank the reviewer for this detail. In fact, the signal to noise ratio does not appear to change dramatically although the phase contrast across the current line it does. Let us assume that the difference in the magnitude of the phase contrast \Delta \phi_2 is mainly due to the difference in the height between the interface position and the MFM tip (see Fig. 1(d)). This apparent constancy of the noise to signal ratio would indicate that the origin of at least part of this “noise” has its origin in the same interface. We have included this detail in the manuscript (second paragraph in Sect. 4.2, lines 233-239).

Reviewer wrote 2) In my opinion, the units in the designation of the y-axes in Figures 3(b), 4(c), 4(d), 5(c), 7(a) and 7(b) should be changed from the unconventional "o" symbol to "deg".

Reply: Thanks. We have corrected it.

Reviewer wrote:  3) The values of the A parameter are presented in a wrong way. Thus, instead of A = 10.5 ± 0.45 × 10-20 (line 211), an appropriate way is A = (1.05 ± 0.05) × 10-19. This kind of unconventional number presentation is found in lines 211, 223 and 224, and two times in the caption to Figure 4.

Reply: Thanks. We have corrected it.

Reviewer wrote:  4) In line 164 in the indication of the sample size a sign "×" is missing.

Reply: Thanks. It is corrected.

Reviewer 2 Report

The search for the room-T superconductivity is an important task. There were many hints that the room-T superconductivity takes place in some islands of graphite materials in the presence of the stacking faults. This possible granular superconductivity was attributed to the existence of the regions with the dispersionless spectrum (the flat band), where the large density of states may highly increase the transition temperature. Later such flat band superconductivity has been observed in twisted bilayer graphene, where the transition temperature highly exceeds the limit dictated by the conventional BCS theory. This confirmed the original idea of the flat band superconductivity in the graphite islands, and the identification of these islands became an important task. The paper describes the new step in this direction, and I recommend it for publication.

Author Response

Reply: We thank the reviewer for this positive comment on our manuscript. We hope that the future readers will also recognize the importance of our experimental work.

Reviewer 3 Report

In this manuscript, the authors studied the granular superconductivity due to stacking faults in natural graphite samples by identifying the flux pinning. Furthermore, by using magnetic force microscopy, the authors claim that similar measurements at lower temperature will be necessary to measure a trapped flux due to the intensity of the signal. Moreover, the authors demonstrate that natural graphite crystals are better ones in order to observe room temperature granular superconductivity. Overall, I think that the paper is well written and organized. The conclusion is evidenced by experimental results. I would suggest publication after the authors can address my following concerns:

  1. Line 80: Can the authors determine what type of stacking faults is seen in the sample shown in Figure 1b? Does different stacking fault affect the granular superconductivity? If this has been studied by someone else, references should better be given here.
  2. Please include standard deviation for Figure 2 if possible.
  3. The authors claimed granular superconductivity 300 K. Except the hysteresis in resistance, is there any other evidence for this, for example, heat capacity? Is it possible to measure heat capacity for such phenomenon? It would be great if the authors can clarify this.

Author Response

Reply: We thank the reviewer for this positive comment on our manuscript.

Reviewer wrote:  Line 80: Can the authors determine what type of stacking faults is seen in the sample shown in Figure 1b?

Reply:  Unfortunately, not. We can distinguish each single crystalline region by its gray color shade due to the different electron diffraction parallel to the graphene planes. The different grey shades originate from a twist around the c-axis or different stacking orders. Most of the single crystalline regions, especially the thicker ones correspond to Bernal stacking order. Due to the much smaller amount of the rhombohedral stacking order, one expects that the thinnest shade regions may correspond to this order. However, any determination of the twist angle or the stacking order via STEM needs an appropriate calibrated sample, which is not available. We included in the manuscript this detail at the lines 88-91.  

Reviewer wrote:  Does different stacking fault affect the granular superconductivity?

Reply: We should differentiate a bit. For example, taking different twist angles between the same Bernal stacking order can have an influence in the macroscopic Tc one measures but also can affect the granular superconducting response. This is expected in case the flat band regions are localized in space and their separation would depend on the characteristics of the SF, e.g., the twist angle. A systematic study has not been done yet, simply because to have a homogeneous SF (without lattice defects) all over a large sample remains a technical problem. Not only the SF is rather difficult (or probably impossible nowadays) to produce artificially but also one needs well-defined crystalline order below and above the SF, if one wants SFs with more than two-layer graphene, without defects in a relatively large area.

Reviewer wrote:  If this has been studied by someone else, references should better be given here.

Reply: The literature shows that twisted two- and three-layer graphene can show different Tc’s or even a single rhombohedral stacking order (clearly below 1K). However, the reviewer is asking about the granular superconductivity issue and this was not studied systematically yet.

Reviewer wrote:  Please include standard deviation for Figure 2 if possible.

Reply: The standard deviation of each resistance point in Fig.2 is less than 2 x 10^-4 R(0). We included it in the corresponding figure caption.

Reviewer wrote:  The authors claimed granular superconductivity 300 K. Except the hysteresis in resistance, is there any other evidence for this, for example, heat capacity? Is it possible to measure heat capacity for such phenomenon? It would be great if the authors can clarify this.

Reply: Granular superconductivity behavior has been measured through I-V characteristics (see new Ref. 27 for example) or magnetization measurements (for HTSC see Ref. 41, and for graphite Ref. 22).

Regarding specific heat measurements as evidence for superconductivity in our samples: they would be rather difficult to achieve. The reason is simply the superconducting mass due to the SFs. Let us assume than we have a sample with 200nm thickness and 1mm^2 area. Within this sample, we have distributed ten superconducting SFs in the whole area. It means that the superconducting mass relative to the total sample (assuming a thickness of ~ 0.5nm per SF) would be in the best-case 5/200 = 1/40. At high temperatures, the phonon heat capacity overwhelms by more than two orders of magnitude the expected electronic contribution. It means that we would need a relative resolution in heat capacity of at least 2 x 10-4 of the background to measure something at Tc.